# Sustainable Development of Employee Lifecycle Management in the Age of Global Challenges: Evidence from China, Russia, and Indonesia

Hua Xiang [1,*], Jie Lu [1], Mikhail E. Kosov [2,3,4], Maria V. Volkova [5], Vadim V. Ponkratov [6], Andrey I. Masterov [6], Izabella D. Elyakova [7], Sergey Yu. Popkov [8], Denis Yu. Taburov [9], Natalia V. Lazareva [10], Iskandar Muda [11], Marina V. Vasiljeva [12,13,*] and Angelina O. Zekiy [14]

[1]  School of Labor Economics, Capital University of Economics and Business, Beijing 100070, China
[2]  Department of State and Municipal Finance, Plekhanov Russian University of Economics, 117997 Moscow, Russia
[3]  Department of Public Finances, Financial University under the Government of the Russian Federation, 125993 Moscow, Russia
[4]  Legal Management Institute HSLA, HSE University, 101000 Moscow, Russia
[5]  Department of Industrial Logistics, Bauman Moscow State Technical University, 105005 Moscow, Russia
[6]  Center of Financial Policy, Financial University under the Government of the Russian Federation, 125993 Moscow, Russia
[7]  Department of Finance and Banking, North-Eastern Federal University, 677010 Yakutsk, Russia
[8]  Department of Financial Management and Financial Law, Moscow Metropolitan Governance Yury Luzhkov University, 107045 Moscow, Russia
[9]  Center for Branch Economics, Research Financial Institute of the Ministry of Finance of the Russian Federation, 127006 Moscow, Russia
[10]  Department of Economics, National University of Science & Technology (MISIS), 119049 Moscow, Russia
[11]  Department of Doctoral Program, Faculty Economic and Business, Universitas Sumatera Utara, Medan 20222, Indonesia
[12]  Top Management, Autonomous Non-Profit Organization "Publishing House Scientific Review" (Nauchnoe Obozrenie), 127051 Moscow, Russia
[13]  Top Management, Atlantic Science and Technology Academic Press, Boston, MA 01233, USA
[14]  Department of Prosthetic Dentistry, I. M. Sechenov First Moscow State Medical University (Sechenov University), 119991 Moscow, Russia
*   Correspondence: xiang.cueb.edu.cn@outlook.com (H.X.); marina.vasiljeva2017@gmail.com (M.V.V.)

**Abstract:** The COVID-19 pandemic has significantly affected the employee lifecycle management (ELM) sphere, leading to the adoption of new human resource (HR) technologies and policies. This study investigates the impact of megatrends, artificial intelligence, digital technologies, and innovation on ELM and human resource management (HRM) policies in China, Russia, and Indonesia. Data were collected through structured interviews and publicly available information from companies in these countries between 2021 and 2022. The study evaluates the effects of artificial intelligence (AI), digital transformation (DT), and innovations on the sustainable development of ELM and identifies differences in technological responses to ELM in companies depending on their level of digital maturity. The results show that the majority of companies have continued the process of ELM digital transformation, but the percentage varies based on the scope of activity, labor, and readiness of the country to implement new technologies. The study reveals that large companies in each analyzed country with over 10,000 employees have a greater need and opportunity to implement HR digital transformation, whereas small companies with up to 100 people can operate without automation. In addition, the findings of this study provide propositions for designing how AI and innovations contribute to ELM. This article contributes to the current debate in the literature by substantiating the positive impact of AI, digital technology, and innovation on ELM and HRM strategies, offering practical applications for companies to improve productivity. Overall, this study highlights the importance of adopting innovative HR technologies in response to global challenges and workplace trends.

**Keywords:** sustainable human resources management; employee lifecycle management(ELM); global challenges; innovation; artificial intelligence; digital technologies; Russia; China; Indonesia; HR digital transformation; HR-Tech

## 1. Introduction

Today, HR-Tech is increasingly penetrating the work of HR departments in various industries, and this trend does not refer to the notorious pursuit of trends or the desire to reduce the staff. HR specialists have an opportunity to eliminate extra workload and deal with more "human", creative tasks associated with motivating employees, improving working conditions, finding talents, and retaining valuable staff. The results of such transformations can be measured in terms of savings, released time, and human resources in production.

The process of digital transformation in HRM brings some problems and concerns about the occurrence of certain errors. Currently, none of the types of management activities can function without applying modern innovative technologies and artificial intelligence. The digitalization of HR processes was previously perceived as a potential threat to HR professionals, as many were worried that with the introduction of digital technologies in processes, recruiters would be replaced, and software products would replace HR managers. However, today, and especially during the COVID pandemic and under the influence of other global trends in the development of the world, it becomes obvious that technologies are the main 'assistants' of HR specialists, which enable them not only to reduce the time for many work processes in the company, but also to improve their quality.

As projected by the McKinsey global institute, by 2030, approximately 3 to 14% or 75 to 375 million workers will be pushed to look for new jobs worldwide, as advances, automation, and digitalization in technologies, such as artificial intelligence, will dominate organizations [1]. Similarly, the sorts of skills needed by organizations will change, with significant implications for careers open to people [2].

This is a challenge for the contemporary workforce; however, the increasing awareness related to it has by far not been able to translate into sufficient practical responses from corporate managers or policymakers. Public spending on the support and training of the labor force has been falling recently in many member countries of organizations such as the OECD (Organization for Economic Co-operation and Development). Likewise, the development and training budgets are not increasing. Certainly, these budgets are and have remained at increasing risk during cost-cutting measures to their inadequately defined business value; however, the existing realities show that this practice will need changes [3]. Thus, this study focuses on the companies, largely foreign MNEs and private firms, operating in three countries(China, Russia, and Indonesia), as there is increasing economic significance, influence, and impact of megatrends in the contemporary business environment, namely, the response to the scarcity of research linked to HRM in the selected countries. The following research questions were considered for the analysis:

1. How are the actions and thinking affected by megatrends (AI, digital technology, and innovation), especially HRM, in firms operating in various countries?
2. In what way do megatrends impact companies in various countries?
3. How do megatrends in various countries impact important HR sub-functions, including performance management, training and development, recruitment and selection, work organizations, compensation and benefits, and retention?
4. What role can social partners and government play in aiding businesses to highlight and address important issues and disseminate HRM practices and policies from national best practices to more conventional sectors and businesses?

Therefore, our study aimed:

1. To analyze the role of the megatrends in influencing the integration of digital technology, artificial intelligence, and innovation changes in employee lifecycle management in Chinese, Russian, and Indonesian companies;

2. To analyze the impact of digital technology, artificial intelligence, and innovation on ELM in Chinese, Russian, and Indonesian companies;

3. To determine which of the three countries has incorporated the most appropriate technological response to the megatrends;

4. To analyze which country (among China, Russia, and Indonesia) appears well-positioned to address the megatrends with the incorporation of digital technology, artificial intelligence, and innovation in ELM;

5. To substantiate the effective ELM model despite a global megatrend based on the best practices.

The significance of this research is in providing high insight to the organizations as they become efficient and productive in ELM. Organizations that possess improved hiring processes will result in less turnover, better hires, and fewer training and rehiring expenses. Using cognitive hiring bots and smart conversational interfaces, artificial intelligence can assist in hiring the best candidates within organizations. Job-specific inquiries can be asked by recruitment bots, which can also offer feedback, updates, and questions about the next step. Based on the anticipated analysis, artificial intelligence can also help in finding the best candidate. Additionally, AI technology can assist companies in forecasting employees who will give the best performance, stay for a longer period in companies, and obtain promotions for their performance. Unlike artificial intelligence, innovation also helps companies make better decisions by generating new ideas and delivering and implementing them within and across the organizations. Therefore, this research paper is a great way to assist companies in developing a perfect ELM model through innovation and artificial intelligence.

## 2. Literature Review

Different factors, such as the emergence of new technologies, social trends, focusing on the importance of environmental issues, etc., have imposed new challenges on HRM in organizations. Since this article aims to investigate the global challenges in this field, in this section of the article, the literature on major megatrends affecting the HRM function, digital technologies, environmental and climate changes, demographic shifts, and globalization and how they affect HR are discussed. At the end of this section, the literature related to ELM is also presented.

### 2.1. Major Megatrends Affecting the HRM Function

An analysis of the relevant extant literature shows that the major megatrends that have affected the HR function, particularly employee lifecycle management, are digitalization, the COVID-19 pandemic, and demographic shifts. Digitalization has affected the way the entire human resource is acquired and managed in organizations. As per Halid et al. [4], rapid digitalization has put increased pressure on HRM to incorporate advanced technologies, such as innovative digital technologies, artificial intelligence, data analytics, etc., to simplify the complex and repetitive HRM processes to keep up with the technological advancements in the industry and emerge as a contemporary origination. The authors have also stressed that innovations need to be brought in by HR professionals to effectively attract, recruit, and retain employees in this digital era. Thus, HRM must adapt to the technological changes brought about by rising digitalization. Démeijer [5] emphasizes that the HRM process has become simpler, easier, and faster due to the rapid introduction of digitalization in the HRM domain. This has enabled HR professionals to save time, make work efficient, and focus more on essential tasks. With respect to recruitment in particular, Nooruddin [6] has suggested that the integration of digitalization in HRM practices has reduced the amount of work required to source suitable candidates analyze skills related to candidate selection and requirements to fill vacancies, and make the final selection. This

not only saves time but also improves the overall ability of the organizations to effectively manage their employees. According to Murphy [7], by enabling workers to work remotely, the use of digital tools can reduce administrative burden, reduce stress, and increase overall employee flexibility and satisfaction with the job and workplace.

Globalization has also influenced the incorporation of digital technologies, artificial intelligence, and increased innovation in employee lifecycle management. It is asserted that amid the rising globalization, staff mobility increased across countries. This trend has influenced companies to incorporate new models of employee management that are more suitable for the globalized economy. Under the influence of globalization, human capital has become more volatile; thus, it requires technological solutions for affective management [8]. The use of digital technology in HRM transforms almost every aspect of human resource management systems, from recruitment to retention in a globalized context [9]. As per Pimenova [10], the increased globalization that has facilitated the global mobility of the staff is the key indicator of advanced digitalization in employee management. Thus, in this global economy, work is no longer tied to one physical location. Rather, work has become digital and virtual, whereby the employees are given the flexibility to work remotely, promoting the concept of a "digital workplace" and the retention of employees by providing them with greater flexibility to work. Thus, organizations are perceived positively by most workers who value flexible working arrangements, thereby contributing to employee retention [8]. These technological changes in the context of HRM and employee management were accelerated during the outbreak of the global pandemic of COVID-19.

According to [10], employees and physical labor are expected to be replaced by technological systems in organizations. The tasks are becoming more automated to require less physical labor. Thus, future HRM and employee management appear to be focused on using artificial intelligence to perform HRM functions, thereby increasing the speed and efficiency of the process. The outbreak of the pandemic has increased the adoption of remote working, the replacement of paperwork and documentation by automated tasks, and various other changes addressing the digitalization and virtualization resulting from the pandemic. In the post-COVID world, the employee structures and HRM have been transformed with increasingly digitalized tasks, whereby employees have become more accustomed to the new working style [11]. Finally, demographic shifts have also led to technological changes in the way employees and workers themselves are managed in organizations. Notably, the younger population are digital natives characterized by the ubiquitous use of digital tools and technologies. Because of this, HRM must incorporate advanced digital technologies to adapt to the behavior of the younger generation and to effectively attract, recruit, motivate, and retain them in organizations. Thus, the upcoming generation is attracted to more technologically forward organizations [12].

### 2.2. Digital Technologies

It is important for companies that seek to maintain a stable position and competitiveness to strive for their own digitalized space. If companies lack financial resources to develop internal HR systems, several ready-made solutions are offered that can be taken as the basis for the digital transformation.

Digital technologies are mainly used in digital computing devices, primarily in computers, in measuring instruments, and in various fields of electrical engineering, such as robotics [13]. The development of digital technologies has made it possible for individual digital devices to be able to process digital information using a microprocessor, perform calculations based on this information, and even make decisions based on the results obtained. However, the emergence of telecommunications, Internet networks, and automation capabilities is also determined by the development and spread of digital technologies [14,15]. Digital technologies should be understood as any electronic technologies that, by deciphering information using a binary code and visualizing it, allow the user to receive information converted into a digital form and perform certain work with it.

Digital technologies as a business solution to improve the HR function efficiency can be interpreted as a certain way of practical application of individual digitization results for searching, organizing, distributing, processing, and storing digital data in the process of implementing the HR function [16].

Currently, new advanced technologies such as big data analytics, virtual and augmented reality technologies, artificial intelligence and its technologies, gamification, cloud technologies, and software products have become available to businesses recently, as these technologies have become cheaper, the computing and functional capabilities of software products have expanded, and the availability of high-speed data transfer has grown, etc. [17].

The above technologies are now strongly influencing the digital transformation of the HR industry, and they are also key tools in the digital transformation. Artificial intelligence and its technologies are gaining popularity. Thus, artificial intelligence (AI) is the ability of systems to imitate human behavior using technologies such as computer vision, natural language processing (chatbot services), speech analytics, decision-making, and recommender systems to perform narrow and specific tasks and gradually learn using the information collected [18,19]. In our opinion, due to the introduction of artificial intelligence and the digitization of the process as a whole, it is possible to solve many urgent problems of developing employee lifecycle management in the age of global challenges.

Researchers, such as Tambe et al. [20], examined the implications for corporate HRM regarding artificial intelligence and found a considerable gap between the reality and promise of artificial intelligence in the domain of HRM. The researchers found significant challenges and problems using data science techniques in human resources, the complication of HR phenomena, accountability problems linked with fairness and legal and ethical limitations, the limitation of data sets, and the possible negative reaction of employees to decisions of managers taken in the context of data-based algorithms. Cappelli et al. [21] highlight increasing concern related to the use of algorithms of artificial intelligence for recruitment. Usually, such algorithms are discriminatory and expensive, and they can end up making wrong decisions about the best candidates. Principally, the technology-linked trends impact several HR sub-functions. However, they mostly influence selection and recruitment, work organizations, training, and development.

A significant trend that has increased as a consequence of the recent COVID-19 pandemic is certainly in increased digitalization of the industrial chain globally and business practices such as teleworking [22]. The outlook and functioning of the global industrial chain are transforming due to digital technology. Among other phenomena, research and development (R&D), medical care, e-commerce, smart functioning, education, delivery of financial service online, contactless distribution, e-commerce, and conduct of online exhibitions have expanded and increased the speeds of domestic and international business and made goods and services globally accessible and integrated. Such changes and developments present significant prospects along with challenges in the context of HRM. Whereas companies are dedicatedly embracing digital technology and artificial intelligence in different business areas, their comprehension of such innovations affecting the employees usually lags behind or is not considered among the priorities.

### 2.3. Environmental and Climate Changes

Environmental degradation and climate change have increased in recent decades, as is seen among global challenges constituting more megatrends [23] with significant implications for practices and policies of management. Organizations have been part of the issues and of the solution to the problems and challenges they are facing [24]. Nevertheless, a review by Jackson et al. [24] related to the existing challenges encountered by HR professionals in North America includes the absence of attention given to long-term climate change implications and environmental degradation, irrespective of the increasing body of evidence proposing that enhancing their environmental performance can positively benefit businesses and organizations. Different opportunities for organizations to engage in

this context are noted in areas such as selection and recruitment (candidates are focusing on the environmental repute of their employers or considered organizations for jobs), training (enhancing the awareness of employees regarding the environmental objective of the organizations), performance management (understanding different environmental activity aspects), and reimbursement (giving incentives and monetary benefits for accomplishing environmental objectives). These challenges are discussed in a number of studies, in addition to [25].

### 2.4. Demographic Shifts

It is observed that demographic shifts are also among the megatrends that carry significant implications for organizations and societies, in relation to client bases and for managing their workforce, especially in organizations. Demographic changes happen in different ways, as highlighted by the United Nations [26]. For instance, population aging is one, and hence workforce aging is also increasing in countries such as China (lesser extent), Japan, and Germany. Elderly workers are required to be unskilled and reskilled to match the new products and technological needs. This trend is synchronized with the increasing youth population in developing countries of Africa and in Vietnam, India, Brazil, and others [26].

Human capital development in emerging economies failed to catch up with the speed of economic development. The migration of talent from South Asian and African countries to the Middle East and Europe and to Pacific Island countries such as New Zealand and Australia, from Southeast Asia, for instance, has increased the skills blockage in the countries of origin, as reported by Tarique [27]. Meanwhile, migration within and between developing countries has been an important facilitator of labor mobility, and more organizations, including MNEs, have relied and are relying on the migrant workforce to fill their positions (skilled and semi-skilled). It is observed that undocumented migrant workers in South Africa usually have lesser employee rights and encounter less supportive conditions and terms of employment, raising questions of equality and fairness. Such discrimination undermines the opportunities for growth and development of migrant workers as businesses globally are faced with the scarcity of skilled labor, according to the International Labor Organization (ILO) and the International Organization of Employers (IOE) [28].

Likewise, gender inequality is also a constant challenge. A rough projection by the IOE and ILO [28] is that 865 million women globally can add to national development and the global economy. Increasing evidence of expansion and growth in employing females exists. Thus, although a recent trend may be found in developing countries, where the participation of females in education is rising, progress is seen in terms of gender equality. Nevertheless, gender-based discrimination continues to be a persistent kind of inequality in almost all countries, as highlighted in a recent report published by the United Nations [29]. Therefore, organizations must review their HR practices and policies, including training and development, working time, work organization, recruitment, and selection, to fix the needs of male and female employees, as this will allow organizations to make good use of human capital and remain connected to the moral imperative of increasing gender equality.

Overall, the global mobility of the workforce and demographic change confront organizations with opportunities and challenges regarding the management of workforce diversity. These trends carry implications linked to language, migration status, ethnicity, gender, age, and culture.

### 2.5. Globalization

In the past decades, economic globalization has become a key feature because economies have increasingly interrelated with each other through integrated global supply chains. Its significance and consequences have been debated for a long time. Nevertheless, recent political and economic developments have confronted several conventional beliefs that appear to have formed a consensus in the recent literature. It is important to observe two

different trends: the first one is the change in the center of gravity from the developed and industrialized economies of the West to the developing and emerging ones. The aggregate GDP (gross domestic product) of G7 countries was approximately USD34.1 trillion in 2015, while that of E7 countries was around USD 18.8 trillion. By 2050, it is projected that the overall GDP of G7 countries is likely to be around USD69.3 trillion, compared to that of E7 countries, which will be USD138.2 trillion [30]. According to Ghauri et al. [31], western countries will hence no longer be at the epicenter of economic globalization.

Additionally, there is increasing political and economic significance of the non-OECD countries, whose organizations, especially MNEs, have unconventionally been in the emphasis of discussions on globalization. The MNEs, for instance, from countries such as China and the Russian Federation, being BRICS members, have faced considerable international economic expansion.

Another noticeable trend is emerging; however, instead of strong de-globalization forces and the relocation of FDI and production activities from one country or region to another or in the same region, MNEs pursue a reduction in the risk themselves from overdependence in particular regions and countries. The initial closure of factories, for instance, undertaken in China between January and March 2020 to avoid the pandemic from spreading led to global shortages of raw materials, equipment, and parts. This had a similar domino impact on the economies of Japan, the United States, European countries, and the Republic of Korea [31]. According to Cooke [32], few organizations in the automotive industry, for the time being, have suspended their production elsewhere apart from China due to the shortage of parts. When production resumed in Chinese factories and several export-oriented firms, laid-off workers were found after reopening factories due to the cancelation of their orders. A sizeable number of apparel factories in Myanmar was closed between February and March, even before the spread of COVID-19 in 2020, due to shipment delays from China. This increased the unemployment of thousands of workers, a significant majority of whom had no social security. Additionally, few governments have seized the prospect of increasing their MNEs by relocating their business back to their home country or relocating their plants and manufacturing units from China to some Asian countries as a strategy to reduce risk. Nevertheless, as reported by Goldthau and Hughes [33], the incentives given by the government through economic packages and directed restoring or bringing home manufacturing will not work as networks of investment and cross-border trade keep costs down and increase innovation and learning. The trends in anti-globalization appear to continue in the ascendant, with the encouraging effect of globalization being probed by both the political right and the left.

The COVID-19 pandemic, along with digital technologies and innovation, has posed significant challenges to HRM, and this, as a consequence, has increased the need for a new set of skills for the workforce, at the same time affecting the way tasks were and are conducted and increasing the platform economy and platform workforces [34,35].

The most crucial investment in companies is their workforce. Recruiting high talent can help companies reach their goals. The employee lifecycle model helps recognize and articulate the different and crucial stages experienced by employees when interacting with their organizations. The employee lifecycle comprises six different stages. Each stage is completed by an employee in order. Human resource's responsibility is to ensure that all employees' requirements are fulfilled to succeed [36]. The first stage is attraction: companies must differentiate themselves from the competition to attract top talents. They should promote good qualities, which can distinguish them from one another in terms of culture and rewards; this measure can make organizations stand out as desirable places to work. The second stage is recruitment, where the best candidates are selected; the organizations must improve their hiring procedures and guarantee that they make the proper choice the first time. The third is onboarding, where the organizations welcome new employees on board and provide significant training required to work with them. The fourth is development, where the companies create satisfying work environments and keep their employees engaged, which will increase employee retention and be helpful

in saving the companies' time and money as it costs time and money to hire new and top talents. The fifth stage focuses on employee retention in the organizations. For this, companies must be transparent in terms of loyalty and trust and should provide employees with flexibility related to work. Companies must recognize employees' hard work and provide rewards to them. The last is separation, where the companies support the decision of employees to leave the organizations to pursue opportunities. Companies must celebrate the contributions made by employees to them throughout the journey [37]. Companies with comprehensive and extensive HR practices and systems (for example, career management, performance-based rewards, internal communication, training and development, performance appraisal, recruitment, and selection) are likely to have employees that are committed, competent, motivated, involved, empowered, flexible, and innovative [38]. These processes can be managed effectively with the help of innovations, digital technologies, and artificial intelligence [39].

Organizations can make their procedures and workflow efficient and productive with the help of technology. The most difficult processes and tasks, such as paying workers, storing data with the help of a cloud server, and a chatbot on the internet, etc., can be made easier and simpler by applying advanced technologies. Artificial intelligence is the most popular technology nowadays and is used worldwide. The employee lifecycle is also managed by artificial intelligence on a great scope [40]. As a result, the companies become competitive and achieve their organizational goals more efficiently as there will be a lower employee turnover rate. For instance, in China, the index level of digital technology has increased from 0.286 to 0.359 from 2001 to 2014, implying that digital technology is paving the way in industries such as manufacturing and organizations [41]. Studies such as that of Devyania et al. [42] have reported that digital technologies, along with artificial intelligence and innovation, are considered in Chinese organizations, especially during the COVID-19 pandemic, for HRM practices, including employee recruitment and training. They use artificial intelligence tools in the recruitment and selection process, in retention, and in employee termination. Likewise, in the context of Russia, the application of digital technologies and artificial intelligence is found to show a positive impact on the development of employee lifecycle management, aiding organizations' HMR practices, such as personnel selection, recruitment, development, and retention [43,44]. The evidence from Indonesian organizations does not differ, as it is found that the emergence of artificial-intelligence-based systems and digital technologies has changed the HR process, including recruitment and selection. The impact of digital technology is found to be positive in terms of facilitating companies to speed up the process of HRM by reducing the cost and time and by recruiting potentially skilled employees from around the world [45].

## 2.6. Employee Lifecycle Management

The term employee lifecycle management model is a human resource model that predicts the overall performance of the employees in certain organizations [46]. In other words, this model is designed to consider every phase of an employee's experience to transform that progression into a structure that offers a unique engagement approach at each level for the employee [47]. This model comprises six different stages, including the processes of attraction, recruitment, onboarding or training, development of the employee, retention, and lastly, separation. In the 21st century, employee engagement strategies are becoming a major part of the business model of talent development projects [48]. This model allows companies to ensure that the employees perform well at different stages. Hence, this model gives most companies a competitive advantage over their rivals. In the employee engagement model, the term engagement is used to illustrate the concept of performance in a way that the employees reflect themselves intellectually, physically, and emotionally through their performance. This is why the employee engagement model proves to be beneficial for employees, as they tend to show a positive attitude and seem to possess learning behavior through these engagement programs [49].

According to Eldor [50], the concept of employee engagement reflects employees being physically, cognitively, and emotionally present at work. However, the employee engagement process is not just confined to the psychological factors of the employees. It also includes the employees' response in reaction to the working environments or physical considerations of the organizations. According to Bakker and Demerouti [51], factors such as the organizational environment and relations with co-workers or superiors also significantly impact employees' performance. Hence, organizations should focus on building a framework that provides a carefully planned experience to the workers to maintain high motivation levels among the workforce. These experiences can be provided easily if organizations deploy technologies such as artificial intelligence into their operations, which would help the companies evaluate the gaps where they must increase employee engagement. This deployment of such technologies can help organizations have full access to all of a particular employee's documents from a single access file of the entire employee lifecycle from "hire to retire" [9].

The employee engagement model, however, believes that an employee's engagement depends on their trait levels or state level, which may vary from person to person. This trait-based engagement model concerns the point of engagement as an issue of innate disposition to illustrate why a person may be engaged at work. In contrast, the state-based engagement model believes that a person's engagement varies according to the experience they face at work, which is inconsistent. To elaborate, the state-based engagement model [52] proclaims that job resources and job demand are the two main factors affecting employee engagement and motivation. The job demand factor includes work pressure in organizations or workload on employees. In comparison, the job resource factors include the physical or physiological aspects that impact employment growth or other abilities such as decision-making power or performance [21].

The advancements in technology in the 21st century in the form of machine learning and artificial intelligence have made task automation significantly easier. Various countries worldwide are competing for supremacy in artificial intelligence, among which China and the US have possessed a widespread adoption of artificial intelligence in their business operations [21]. China solely bases its operation on the human and technological workforce to sustain the economic dominance it possesses worldwide, which is why China is leading the digital economy. These digital technologies, specifically artificial intelligence, have proved to improve the production of goods and services by enormous growth, making companies and industries in China compete globally.

As exemplified by some of the biggest companies in China that are incorporating artificial intelligence in their operations, such as Alibaba, Baidu, etc., studies have shown a positive impact on the integration of artificial intelligence in human resource operations [53]. It fosters the employee lifecycle, as artificial intelligence facilitates the recruitment process of the employees, giving an unbiased selection and speeding up operations. Moreover, the integration of artificial intelligence can lead to the ease of many automation processes in administrative tasks, some of which include the screening of candidate applications or scheduling interviews. Additionally, artificial intelligence can help improve employee retention within the organizations, sincehuman resource management, through these technological advancements, can now gauge employee performance better than ever, which enables the HR department to provide employee feedback resulting in high employee motivation [54]. All leading world economies, including Russia and China, highly believe in these technological advancements, and this serves a reason why they are leading economies.

All companies in China have integrated or are looking forward to integrating artificial intelligence into their human resource practices. When it comes to the application of artificial intelligence incorporation in HRM practices, human resource departments also have to face different challenges. For example, deploying certain algorithms in recruitment is difficult as those algorithms do not possess the sense of decision-making of who to hire. Research has shown that Russian leaders or managers in Russian companies have also expressed concern about implementing artificial intelligence in HR practices, especially as

AI lacks the capacity for emotional intelligence, which can impede many HR processes [47]. However, HR departments in different organizations across the world have incorporated these intelligent technologies, considering the benefits they provide and not neglecting the fact that they are a way to gain a competitive advantage in today's world. According to Nikpour and Semushkina [55], there is a direct relationship between the companies' productivity and the use of artificial intelligence as it speeds up many operations while performing them efficiently. Hence, the integration of these models based on advanced technology can enhance employee lifecycle management in organizations.

## 3. Research Methods

This study emphasizes the employee lifecycle management analysis in a dynamic innovation and artificial intelligence context. Therefore, to accomplish the aim of the research, these chapters are significant in discussing the methods that can help achieve the objectives. This chapter is categorized into sub-sections related to research methods, including philosophy, approach, and research design. Moreover, it provides details about the data collection process, including the sample size and techniques for data analysis. In the end, the ethical consideration is also included to reflect that this research must abide by ethical rules and conduct. The flowchart of the research methodology is represented in the Figure 1.



**Figure 1.** The approach of this study to achieve the research objectives.

The philosophy of this research is associated with beliefs about the view of the world and enables knowledge creation [55]. Research philosophies can be determined on the basis of the methods, ontology, and epistemology and include positivist, interpretivism, pragmatic, and realism philosophies. However, the selection of research philosophy is related to the adoption of the research approach and design. Considering the aim of this study, an interpretivism philosophy is adopted, which appears suitable as it relies on addressing the research problem, assessing multiple realities, and relying on the use of qualitative methods, which are required in this study. This philosophy does not rely on statistical tools for evaluation, which is in line with qualitative research. In contrast, the positivist philosophy relies on a single reality, depends on credible facts and data, and focuses on a quantitative approach to addressing the research gap. This philosophy is measured using structured methods and processes and relies on testing hypotheses; hence, it is not considered for this study because the study does not demand testing a hypothesis or evaluating facts using a quantitative approach. Moreover, this philosophy is only useful in analyzing data and relies on human participation as the investigator has less control to influence the process of data collection [56]. Therefore, interpretivism is deemed appropriate in providing comprehensive and in-depth details linked to employee lifecycle management in the innovation and AI context. Precisely, the interpretivism philosophy is a suitable philosophy that can aid in exploring multifaceted and complex issues such as employee lifecycle management. By focusing on the individual experience of employees, the considered philosophy can also help identify specific opportunities and challenges that arise in the context of innovation and artificial intelligence.

Similar to the philosophies, the research approach is an important part of the methodology and is divided into qualitative, quantitative, and mixed-method approaches. The selection of the research approach complies with the assumed research philosophy; there-

fore, for this study, a qualitative approach is suitable to investigate the role of artificial intelligence and innovation in managing the employees' lifestyle and to analyze the development of the lifecycle management model in the context of artificial intelligence. The qualitative approach is significant in presenting a broad and critical analysis of the research problem, unlike the quantitative approach, which emphasizes statistical analysis to measure the data using different techniques, such as computational and statistical, and generates causality statements [57,58]. The use of a quantitative approach may not be the best option for analyzing employee lifecycle management in the context of innovation and artificial intelligence. Employee lifecycle management comprises a range of multifaceted and complex issues that may not be captured easily through numerical data alone. For instance, understanding the impact of artificial intelligence and innovation on employees may need an in-depth exploration of their subjective experiences, perception, and attitude. These types of data are not easily quantifiable and may need a qualitative approach. Therefore, the qualitative approach is useful and deemed suitable for a comprehensive analysis of data using non-statistical tools. Moreover, it is compatible with the assumed research philosophy. Therefore, a qualitative approach is important to achieve these objectives. Moreover, with the help of a qualitative approach, in-depth and comprehensive data linked to employee lifecycle management innovation and AI can be acquired on the basis of the evidence from China, Russia, and Indonesia; hence, the selection of this approach is justified.

The entire strategy for conducting research, ranging from data collection, instruments, and analysis, is referred to as the research design, which can be classified into different forms such as reviewed design, experimental, semi-experimental, meta-analysis, descriptive, and correlation design, etc. However, their use depends on the objective and purpose of the study [59]. In line with the assumed research philosophy and research design, an explanatory design is considered, which is a precursor to qualitative studies for examining the research problem that had not been explored previously [59]. The phenomenon of developing model employee lifecycle management in the innovation and AI dynamic context with evidence from Russia, Indonesia, and China has not been explored in previous studies. Additionally, an explanatory design is significantly useful and is correctly applied in this study to help identify the types of employees who are likely to benefit from a training program or the specific organizational context in which specific HR practices are effective. Therefore, using an explanatory design, the researchers provide details about the constructs and focus on the reasons for selecting an object of the study. Other designs, such as correlation and descriptive ones, are not suitable as they are used mainly in quantitative studies. Moreover, the explanatory design is compatible with the overall objectives of this study and is relevant in presenting unexplored data linked to the current research area.

To collect data, primary or secondary sources are used depending on the requirement of the study and the selection of the research design. In this study, the primary source is selected as it helps in retrieving data from the participants who are new and recent, unlike secondary sources, where the data is already available and published in the form of articles and journals [59]. In terms of the data collection process, a structured online interview is selected to derive information from the participants (employees of tech companies), including top managers of human resource departments, heads of departments of human resources management, heads of departments of information technology managers, and recruitment managers from Russia (260 people from 15 firms), Indonesia (425 people from 15 firms), and China (550 people from 25 firms). The reason behind selecting structured interviews is that they allowed in-depth exploration of employees' subjective experiences and perspectives and provided a more comprehensive and nuanced understanding of their perspectives. The 1235 participants were contacted through social networking websites, i.e., LinkedIn, where their contact information was taken. A formal request was sent to them through email to give consent for this study. With their approval and signature on a consent form, the process of data collection started with a structured interview employing a questionnaire that included free-form questions (Appendix A). A short form was sent to the participants to fill in their personal details and experience before proceeding with the

interviews. The participants were contacted for the data collection, and 1500 participants were considered initially for the interviews. However, with regard to the time limitations and availability of the participants, a total sample of 1235 respondents was finalized. Nevertheless, this sample size is adequate to accomplish the research objectives; however, the findings will be insufficient to be generalized in a larger context. The participants were recruited based on anon-probability convenience sampling strategy in which the participants were selected and contacted due to their accessibility, availability, and ease of use [58]. The overall duration of the data collection process lasted around 2 weeks, with both interviews lasting around 40 min. The interviews were conducted based on the availability of the researchers; therefore, the duration of data collection was long due to the strict working schedules of the participants.

Since quantifying the number of digital human resources managers is difficult, accurately describing the digital human resources managers is a challenge. Our study found some commonalities in the profile of digital workforce managers in the three countries studied. They tended to be younger than the total workforce. In Russia, digital workers average about thirty years of age, and they largely represent a younger age group (18–26 years old). In China, the profile of digital workers is similar to Russia. They are, on average, younger in China than in the rest of the world.

The gender distribution is clear. Male platform workers outnumber their female counterparts. On average, only 35 percent of online platform workers in China were women. This gender distribution is similar to the results from other studies. However, female participation rates on platforms in industrialized countries are considered higher than those observed in China. These data suggest that the gender gap has narrowed recently. For example, the proportion of Russian women working on digital platforms rose to 48 percent in 2021. While the share of young women adopting digitalization has increased rapidly in the countries we studied, men still make up the largest share.

To analyze the research data acquired from structured interviews with participants from Russia, Indonesia, and China, the thematic analysis technique was selected to analyze the qualitative data. The technique was applied to analyze texts of this study; however, it is also applied for analyzing audio and videos, which were not required in this study [60]. Overall, the thematic analysis was performed on the basis of key themes associated with the research objectives and was supported by the previous literature to argue whether the findings are associated with previous studies or contradict them. Thematic analysis was deemed suitable as it helped in identifying specific ways in which artificial intelligence and innovation are affecting employees' perceptions and experiences. By identifying the patterns and themes within employees' experiences, the thematic analysis provides a nuanced and in-depth understanding of how these technologies are affecting the workplace. In qualitative studies, the ethical considerations should be emphasized to assess whether ethical rules and principles were followed during the study [61]. Thus, this research carefully followed ethical rules, including informing the research participants about the objective of the study and taking their approval and written consent before starting the data collection process, ensuring and protecting the rights and interests of the participants, their safety, and privacy of the details provided by them. Moreover, the freedom to leave the study was given to the participants, and the data were collected with convenience and in free time, giving space for the participants to freely express their opinions and beliefs.

## 4. Results

This section presents a thematic analysis of the findings obtained in the interviews with 1235 HR managers from the case companies. Despite the existence of some differences in the employee lifecycle management of the tech companies in the three countries, the findings revealed that artificial intelligence, digital technologies, and innovation have transformed the processes. The following subchapters compare how different stages of employee lifecycle management are conducted at the 55 companies.

*4.1. Recruitment and Hiring*

The first stage of employee lifecycle management is recruitment, whereby the most critical task is to identify and select the right individual with qualifications, skills, and competencies matching the job description. Artificial intelligence and digital technologies have played major roles in the recruitment processes of the three countries despite megatrends, such as globalization, digitalization, COVID-19, and demographic shifts. The role of artificial intelligence was closely associated with automating the recruitment and hiring process, whereby the procedures were much more formalized in China than in Indonesia, with Russia being somewhere in between.

Five hundred and fifty Chinese HR managers reported that in their companies, artificial intelligence had been actively used for the recruitment of the appropriate candidates based on the use of the multiple attribute decision-making method (MADM) combined with predictive analytics. The MADM model is used for prioritizing options based on certain criteria (attributes). This aids in evaluating and selecting candidates with the required skills. According to the Chinese HR managers,

> *"We have been using artificial intelligence for both attracting and recruiting employees. This helps us access the global pool of potential candidates and attract the younger population that actively seeks jobs online or has created online profiles. In the attraction stage, the recruitment managers use predictive analytics to find the candidates from the received applications. Once the candidates are shortlisted, they get a call for an interview, and then the recruitment managers use cognitive recruitment bots to recruit the people into the organizations . . . These bots were introduced after the outbreak of the pandemic when the interviews could not be conducted in face-to-face settings and to facilitate the overseas candidates. The bots can have intelligent conversations with the candidates to assess them on the basis of predefined criteria. The bot uses the criteria to assess if the candidate is the right fit for the job. Finally, predictive analytics is also used for predicting the future outcomes of the selected candidates".*

This demonstrates that the HR managers at the Chinese firms have fully automated the recruitment and hiring process as a response to the global megatrends with the incorporation of artificial intelligence in the process. This aligns with the findings of theextant literature as the integration of AI in recruitment transforms the recruitment process by automating various essential tasks such as information checks, resume assessment, and verification of the provided information, etc. [62].

When Russian HR managers were asked how they had been using AI for this stage, 260 of them from 15 firms stated:

> *"The recruitment managers at our companies use artificial intelligence-powered background checks given the plethora of resumes we receive from across the globe. This shortens the whole filtering and recruitment process and makes it easier to recruit people from all around the world, thereby expanding the pool of potential candidates. The shortlisted candidates are called for an interview. At this stage, artificial intelligence is not used directly to conduct the interview but rather as conversation analytics during the interview. It records the entire interview and transcribes it to compare all the responses. Thus, the manager does not need to transcribe each response on his own. Moreover, during COVID, we incorporated tele-interviews to facilitate the recruitment and hiring process. The interviews were conducted remotely, and the conversation analytics recorded and transcribed the interview responses".*

Thus, in Russian firms, the involvement of artificial intelligence in the interview process is rather passive and secondary in comparison to that in Chinese firms. Hence, the first proposition of this study is designed as follows:

**Proposition 1.** *Predictive analytics algorithms are used for finding candidates in the recruitment and hiring processes.*

Indonesian firms are at the lowest end of the continuum, with the lowest incorporation of artificial intelligence and digitalization. Nonetheless, 425 Indonesian HR managers from 15 firms reported the use of machine learning in the recruitment process. One of them stated that the companies use the K-nearest neighbors (KNN) model algorithm for resume screening and ranking. KNN is a machine learning algorithm that is used for predictive and classification problems. The Indonesian manager stated:

*"We use KNN for the ranking of the resumes we receive for our job postings from local and overseas applications. This way, we can identify the top resumes that match our criteria and requirements. The hiring manager then shortlist the top candidates based on the ranking and calls them for an interview".* The rest of the recruitment procedure is conducted manually by the hiring manager. Thus, the use of artificial intelligence is limited to measuring the degree to which a resume appears to be matching the required skills, qualifications, and competencies. Therefore, the second proposition of this study is deducted as follows:

**Proposition 2.** *AI algorithms are used for AI-driven interviews and shortlisting in the recruitment and hiring processes.*

Nonetheless, the responses of the 1235 HR, recruitment, and information technology managers from 15 Russian companies, 25 Chinese companies, and 15 Indonesian companies showed that the adoption of artificial intelligence and digital technologies had transformed the recruitment and hiring process, making it shorter, easier, and cost-effective, which supports the findings of the literature review [37,54].

Figures 2–4 summarize the stages of the recruitment process in the Chinese, Russian, and Indonesian firms.

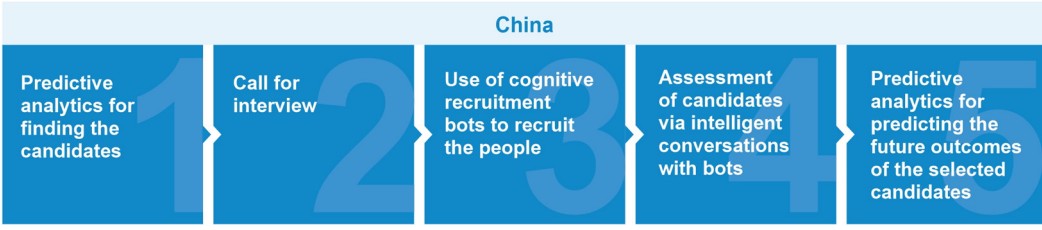

**Figure 2.** Stages of the recruitment process in Chinese firms.

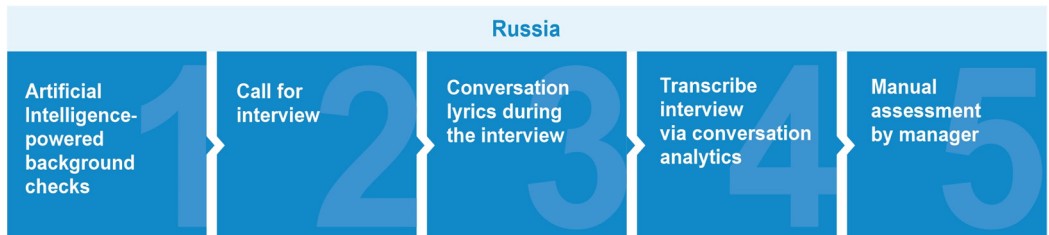

**Figure 3.** Stages of the recruitment process in Russian firms.

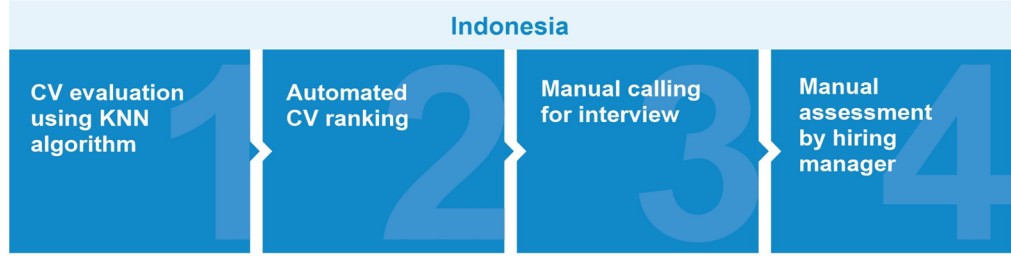

**Figure 4.** Stages of the recruitment process in Indonesian firms.

*4.2. Onboarding*

The incorporation of technology can enable firms to simplify the onboarding process by eliminating or shortening tasks such as paperwork, orientation, and training seminars, etc. This was true in the case of all 55 firms. As per a Chinese manager: *"The use of artificial intelligence has simplified the mundane tasks of onboarding. Vital tasks, such as account generation, providing orientation and training, and document generation are handled by the AI program. The HR manager just has to check if the documents are properly signed. This has been further enhanced after the COVID outbreak as we had to process all the paperwork for the employees that were hired during the pandemic. The AI-powered chatbot has also been trained to answer the frequently asked questions of the employees. Thus, new employees can easily receive all the guidance and answer their queries without the need to approach anyone. The paperwork, thus, can be completed in a remote manner as well".*

Therefore, the third proposition of this study is considered as follows:

**Proposition 3.** *AI-powered online chatbots are used in the onboarding process to answer queries and provide guidance.*

Similarly, a Russian manager asserted, *"AI is being used in the form of automated onboarding software for processing all the paperwork of the new hires. All the new employees are provided with remote access to forms, and they can easily submit their queries and requirements without the need to wait for some employee/manager to have time to answer their queries. The software also sends out surveys at periodic intervals to collect feedback from the employees".*

Consequently, the fourth proposition of this study can be written as follows:

**Proposition 4.** *AI algorithms are used for automated paperwork and documentation in the onboarding process.*

In the case of Indonesian firms, the 425 HR managers (from 15 companies) asserted that during the COVID-19 pandemic, the companies shifted to remote onboarding to adapt to this megatrend and facilitate the continuation of their important employee management procedures. An Indonesian manager stated, *"The onboarding processes, such as account setup and documentation, have become automated. The employees are sent welcome emails and necessary forms to fill out, while an FAQ platform has been set-up for addressing common employee queries. The employees can also submit their queries* via *the company portal".* Thus, the onboarding processes of the companies have been impacted by digitalization, globalization (global mobility of the staff), and COVID-19. AI-based programs have shortened and simplified the companies' onboarding processes by automating mundane and time-consuming tasks. This supports the prevalent idea in the extant literature that the use of AI in onboarding can save time for the HR function by automating the onboarding tasks such as space allocation or profile creation for the new employee, which ultimately enables the HR department to focus their attention on more value-added work [63]. Thus, the fifth proposition of this study is written as follows:

**Proposition 5.** *AI-based programs are used in onboarding for the purposes of orientation, training, and collecting feedback.*

Additionally, the 550 Chinese recruitment managers (from 25 Chinese firms) asserted, *"We use an organizational network analysis (ONA) model for tracking the collaboration among the employees and identifying the influencers and energizers in each department of the firms. These influencers are considered go-to people in the firms. Thus, ONA is used for onboarding of the new employees by connecting them to the influencers within their own business unit".* Thus, this leads to a smoother onboarding of the new employees by fostering engagement among the existing employees and new hires. Therefore, the sixth proposition of this study is formulated as follows:

**Proposition 6.** *Organizational network analysis (ONA) models are used in onboarding to managing employee relationships.*

This result appears to comply with the findings of Ritz et al. [64], according to which automation of onboarding functions such as welcoming new employees via email, connecting new employees with teammates, and personalized messages from HR managers and supervisors boost the confidence of the new employees, make them feel welcomed, and foster engagement and socialization among the employees. The differences in the onboarding process of Chinese, Russian, and Indonesian firms are summarized below (Table 1):

**Table 1.** The onboarding process of 25 Chinese, 15 Russian, and 15 Indonesian companies with a number of employees around 550, 260, and 425.

| Chinese Firms | Russian Firms | Indonesian Firms |
|---|---|---|
| Account generation<br>Orientation and training<br>Document generation<br>AI-powered chatbot to answer frequently asked questions<br>ONA for fostering engagement with new hires | Automated paperwork<br>Remote administration<br>Surveys for employee feedback | Account generation and employee welcome<br>Automated documentation<br>Digital query and support |

*4.3. Development*

The third stage is development, which is concerned with improving and enhancing the extant skills and competencies of the employees, ensuring that they are well-positioned to address the evolving organizational goals and industrial demands. Digital technologies and artificial intelligence have transformed this stage of the employee lifecycle. In this regard, Chinese HR managers asserted that *"managers at our companies use artificial intelligence for the employees' performance management. Artificial intelligence-powered software is being used to develop goals, collect feedback, track progress, and evaluate the employees' performance. The program collects data from multiple sources, such as the supervisors, subordinates, overseas subsidiaries with which employees have worked, customers etc., to rate the employees' performance on the basis of predefined goals and criteria . . . The use of artificial intelligence has also enabled firms to use a continuous performance management system that is based on real-time reviews and progression data. This way, employees can track their performance in real-time rather than getting feedback after intervals/periods . . . The program is also used to predict the future performance of the employees based on their past performance trends".*

Similarly, Russian managers also asserted that *"AI-based software is used for managing the performance of all employees for periodic performance management, and the performance is calculated at regular intervals. The software summarizes the achievements and performance of employees against predefined goals. This way, the manager can keep an accurate track of the KPIs and evaluate the performance of the employees on uniform KPIs".*

Indonesian firms have also incorporated digital technology and artificial intelligence for performance management. As per an Indonesian manager, *"a digital software has been developed based on the 360-Degree Feedback model whereby the employees and managers are asked to anonymously give feedback on the performance of other employees online by filling the provided questionnaire. After this, the role of artificial intelligence comes in whereby the employees are evaluated on the basis of the established criteria, and their weak areas are identified".*

Hence, the seventh proposition of this study can be written as follows:

**Proposition 7.** *AI algorithms are used in the retention and development processes for performance management (i.e., AI-driven goal tacking, continuous performance evaluation and real-time feedback, automated tracking of KPIs, performance prediction).*

These examples of the use of digital technologies and artificial intelligence in performance management show that they have facilitated more data-driven decisions in companies. These performance management strategies are well-supported by the literature as AI can gauge employee performance in a better way than conventionally. The automated provision of performance feedback motivates the employees [54].

Nonetheless, after performance management, artificial intelligence has had an active role in the learning management or training and development of employees. As per Chinese managers, *"the performance management software also identifies the issues pertinent to the employees' performance and builds personalized training and learning programs on the basis of the data. Thus, customized learning programs are developed for the employees by recommending learning material and content"*. The same approach is used in the case of Russian firms. Russian managers asserted, *"Adaptive learning model is used with the adoption of artificial intelligence. Artificial intelligence provides insights to the employee on the managers pertinent to the performance issues and improvement areas. Then, the HR managers can develop customized training and development programs for the employees"*. Thus, the use of AI-based smart technologies can assess the learning needs of the employees, which enables the HR managers to match them with adequate training and learning programs for the employees [65], showing that the use of AI facilitates the development of accurate and most-relevant learning programs. Therefore, the eighth proposition of this study is written as follows:

**Proposition 8.** *AI algorithms are used in the retention and development processes for learning management (i.e., identification of performance issues, development of customized training and learning programs).*

Table 2 summarizes the development stage of employee lifecycle management in the Chinese, Russian, and Indonesian firms.

**Table 2.** Summary of the development stage of employee lifecycle management in 25 Chinese, 15 Russian, and 15 Indonesian firms with a number of employees around 550, 260, and 425.

| Chinese Firms | Russian Firms | Indonesian Firms |
|---|---|---|
| Performance evaluation and management by the automated collection of data from multiple sources and analysis Appraisal Learning management and training Customized learning plans | Performance evaluation and management by assessment of the employee achievement against KPIs Appraisal Learning management and training using an adaptive learning model Customized learning plans | Performance evaluation and management by 360-degree feedback Appraisal |

## 4.4. Retention

In the case of retention, the focus of artificial intelligence and innovation facilitates the management of employee promotions and benefits. In all 55 firms, the results of the automated performance management were used as the basis for identifying opportunities for employee promotions. To manage employee benefits to retrain them, 55 companies have used varying approaches. The 550 Chinese HR managers from 25 companies asserted that they use artificial-intelligence-powered chatbots/programs for servicing employee benefits. These AI-powered chatbots/programs are used to create the best employee benefits package for each employee in the organization. Chinese managers stated, *"we use an Artificial Intelligence-based program and chatbots that enable employees to select and manage their benefits. The chatbot asks various questions regarding the financial standing and preparedness, risk tolerance, and health condition of the employees to make recommendations for the best-fit benefits package for the employees. Chatbots also connect the employees with the best insurance providers"*. Hence, the ninth proposition of this study is considered as follows:

**Proposition 9.** *AI algorithms are used in the retention and development processes for employee benefits management (i.e., facilitating the customized benefits package, connecting with insurance providers).*

According to the findings, Russia and Indonesia lack automated management of employee benefits, demonstrating the leadership position of the Chinese tech firms in this regard. Nonetheless, along with artificial intelligence, innovation focus also has a major role in the retention and development of employees. The 1235 HR managers of these 55 firms from Russia, Indonesia, and China have endorsed the idea. Thus, a manager of one of 25 Chinese firms asserted, *"innovation is being fostered through open communication. We have a platform in the companies whereby the employees are allowed to openly share their ideas and give feedback on the ideas of others. The employees are encouraged to share ideas for the new product development".*

A manager of one of 15 Russian firms reported a somewhat similar scenario. He stated, *"we have a co-working space platform where managers often ask for ideas and suggestions. The managers also tend to have contests among the employees to explore innovative ideas".* In a similar vein, the HR manager of one of the Indonesian firms posited, *"we have an internal corporate social network whereby employees can openly communicate with one another and share information and ideas. The employees often post suggestions or ask for ideas from their coworkers on the platform. This has boosted innovation and employee relationships providing employees with a good working culture that ultimately boosts retention".* These findings align well with the widely agreed role of the open option, which is to accelerate the flow of knowledge within and across boundaries [66,67], which ultimately contributes to increased innovativeness and technological improvement.

When asked about the benefit of this approach in the context of employee lifestyle management, a Chinese manager asserted that *"it has a direct influence on the employee's retention. When employees are given the opportunity to share ideas, they feel valued by their employers. Moreover, such open sharing of information and ideas among the employees fosters greater collaboration and engagement among the employees, fostering better employee relationships. Hence, they are more likely to be retained by the companies as they feel satisfied and happy with the overall culture".* Similarly, a manager of a Russian firm reported, *" . . . using innovation creates a more dynamic culture that not only contributes to employee retention but also to overall employee development".* Finally, an Indonesian manager reported, *"employee retention can be explained by better engagement, collaboration, and the opportunity to share ideas. In terms of development, innovation has improved the employee's mobility as employees can work on different projects. The HR managers tend to select different employees for projects based on their ideas and suggestions".* All in all, innovation plays a role in the retention and development stage of employee lifecycle management.

The use of digital technologies to facilitate a more flexible working environment was also found to be a major initiative taken by Chinese firms to boost employee satisfaction and overall retention. As per Chinese managers, *"online portals and digital technologies have been incorporated after the pandemic to enable the employees to work remotely. Recognizing the effects of this on overall working flexibility and ultimately on employee retention, we continued this approach in the post-pandemic period as well. This way, our employees can manage their work from any part of the world in a flexible manner".* This appears to be in line with the findings of Melnychenko [8], who suggested that digitalization gives employees greater flexibility to work, which contributes to employee retention. Therefore, it can be written that:

**Proposition 10.** *Open innovations such as open platforms contribute to the employee engagement and collaboration phase of retention and the development of ELC, with mechanisms such as enabling sharing of ideas and suggestions and providing opportunities for contests for exploring innovative employees.*

Table 3 summarizes the retention stage of employee lifecycle management in the Chinese, Russian, and Indonesian firms.

**Table 3.** Retention of employees at 25 Chinese, 15 Russian, and 15 Indonesian firms with a number of employees around 550, 260, and 425.

| Chinese Firms | Russian Firms | Indonesian Firms |
| --- | --- | --- |
| Management of employee benefits Knowledge exchange and development to foster innovation and relationships Remote working for flexible working | Knowledge exchange and development to foster innovation and relationships | Knowledge exchange and development to foster innovation and relationships |

*4.5. Exit/Separation*

The last stage of the employee lifecycle is separation, whereby the employees leave the organization. Digital technology and artificial intelligence have played a role at this stage as well. The 55 firms have had varying innovative uses of digital technologies and artificial intelligence at this stage. In the case of 15 Indonesian firms, artificial intelligence is used for managing the employees' final paperwork. An Indonesian manager reported: *"we have a portal for processing all the essential paperwork, including the final paycheck and compensation, healthcare insurance expiration, etc.".* Thus, the use of AI has made the offboarding process easier, shorter, and smoother and has eliminated the need for lengthy paperwork. In the case of 15 Russian firms, it is found that *"the AI is used to send a digital notification to all the departments, such as security, payroll, and admin, about the employee resignation. All the paperwork, such as final pay and insurance plans, are processed in an automated manner, and once the employee leaves, the access to companies' systems and applications is automatically revoked".*

**Proposition 11.** *AI algorithms are used for automated paperwork and documentation in the off-boarding process.*

On the other hand, in the case of 25 Chinese firms, the use of digital tools and artificial intelligence is found to be more comprehensive and innovative than mere paperwork. As per Chinese managers, *"in addition to automated offboarding in terms of paperwork, documentation, and pay finalization, the same intelligent chatbot is used to have an intelligent conversation with the employees that are leaving. The process helps in identifying the satisfaction level of the employees that are leaving and the major reason behind the resigning employees".* Thus, these systems and tools are used to analyze and predict employee attrition in companies.

**Proposition 12.** *AI-powered online chatbots are used in the off-boarding process for exit interviews.*

Table 4 summarizes the exit stage of employee lifecycle management in the Chinese, Russian, and Indonesian firms:

**Table 4.** The exit stage at 25 Chinese, 15 Russian, and 15 Indonesian firms with a number of employees around 550, 260, and 425.

| Chinese Firms | Russian Firms | Indonesian Firms |
| --- | --- | --- |
| Automated documentation and paperwork Pay finalization Predicting employee attrition to reduce turnover in the future | Automated documentation and paperwork Pay finalization Notifying the stakeholders Access revocation | Automated documentation and paperwork Pay finalization |

All in all, these findings indicate that the megatrends of digitalization and global challenges have influenced companies to incorporate more innovative approaches to employee lifecycle management with the application of advanced digital technologies and artificial intelligence tools.

### 4.6. Model for Employee Lifecycle Management Based on Digital Technologies, Artificial Intelligence and Innovation

Despite the differences in ELM in the firms, the findings of interviews in terms of best practices can be combined to create a conceptual model of the cumulative effect for employee lifecycle management in the dynamic context of innovation and artificial intelligence. This study develops a theoretical framework based on 12 propositions resulting from the findings (see Figure 5). It suggests managing the employee lifecycle in a better and seamless way and can be incorporated across a range of sectors in the face of current megatrends.

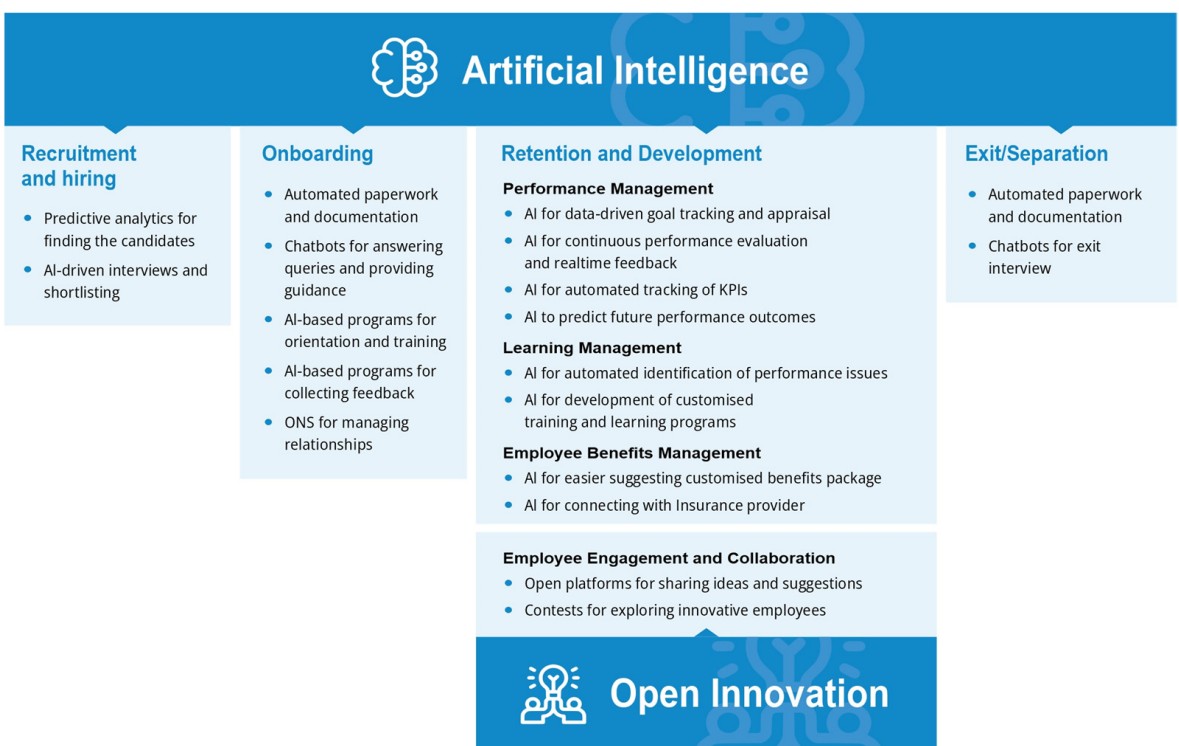

**Figure 5.** A conceptual model of the cumulative effect on employee lifecycle management in the dynamic context of innovation and artificial intelligence.

On the basis of the evidence collected from the companies, the aforementioned model has been proposed. The model demonstrates that the incorporation of artificial intelligence has transformed the entire employee lifecycle from the recruitment stage to exiting the organizations, while the incorporation of innovation has contributed to better employee retention and development. At the stage of recruitment, artificial intelligence contributes to the automated recruitment and hiring process, eliminating the need for human intervention, and making the process easier and more accurate. At the onboarding stage, AI-based programs and tools can be incorporated for smoother paperwork and documentation, for continuous guidance, orientation, and training of the new employees, and to connect the employees with others in the organizations. At the employee retention and development stage, the companies can leverage both artificial intelligence and innovation. Artificial intelligence can facilitate automated performance management, employees' customized learning and training, and automated enrollment and management of their benefits. On the other hand, innovation can be incorporated to foster better engagement and collaboration and make employees feel valued, which ultimately contributes to employee retention. Simultaneously, with the innovation, employees can be given better mobility opportunities that contribute to better employee learning within the organizations. Finally, at the exit stage, the companies can use artificial intelligence for automated paperwork collecting employee feedback, similar to the onboarding stage. The automation and improvement of all of these stages of the employee lifecycle show the verification of the research hypothesis

stating that the adoption of innovation and artificial intelligence facilitates better management of each stage of the employee lifecycle at the companies and has a positive impact on the employee lifecycle management in a dynamic context in China, Russia, and Indonesia.

## 5. Discussion

### 5.1. Comparison of the Findings and Employee Lifecycle Management Model against the Megatrends

Analyzing the findings of the interviews, it can be deduced that 25 Chinese firms participating in the research have incorporated artificial intelligence, digital technologies, and innovation in a more comprehensive manner for managing the employee lifecycle as a response to the megatrends: globalization, digitalization, demographic shifts, and the COVID-19 pandemic. Chinese firms are found to be leveraging artificial intelligence, digital technologies, and innovation from each stage of the employee lifecycle in terms of attracting, recruiting, and onboarding employees, performance management and appraisal learning management, employee benefits management, and resigning employees. On the other hand, the Russian and Indonesian firms were found to have taken a narrower approach toward their incorporation, as some part of the work at each stage of the employee lifecycle was managed manually in conjunction with digital and artificial intelligence tools. Indonesian firms reflected the lowest incorporation of artificial intelligence and digital technologies in their employee lifecycle management.

When it comes to recruitment, all 55 firms have demonstrated the adoption of digital technologies and artificial intelligence and have transformed their recruitment and hiring procedures. The recruitment process of 25 Chinese firms appears to be fully automated and powered by artificial intelligence compared to that of 15 Russian and Indonesian companies, where some stages are conducted manually. Thus, the use of digital technologies and artificial intelligence for recruitment and hiring was found to be more comprehensive in the case of Chinese firms. It is asserted that the automation of the recruitment stage increases the effectiveness and accuracy of the process and makes it less worker-intensive and more cost- and time-efficient for the employers [68]. The recruitment process, with the use of digital technologies and artificial intelligence, facilitated the companies to recruit employees from a global pool, demonstrating global staff mobility. Many changes in recruitment were found to appear after the pandemic outbreak. The Chinese firms also demonstrated that they have intentionally incorporated digital tools for the recruitment of the younger population that increasingly use digital technologies.

Despite the level of automation in their recruitment stage, with the Chinese firms being a step ahead, 25 firms have successfully shortened their overall recruitment process, making it more efficient and convenient for the HR managers. Furthermore, the use of artificial intelligence in terms of predictive analytics facilitates unbiased selection and speeds up the recruitment process. In the case of Chinese firms, it is noted that the firms have fully automated their recruitment process. Thus, the integration of artificial intelligence has led to the ease of many administrative tasks with the automation of processes.

Similarly, in the context of the onboarding stage, it is found that the adoption of digital technologies and artificial intelligence has enabled the firms to simplify the onboarding with the elimination of various tasks. The use of artificial intelligence simplifies and shortens the various procedures such as paperwork, orientation, and training. All 55 firms have been able to make their onboarding time- and resource-efficient, which allows for the processing of all paperwork remotely and demonstrates the impact of both globalization and digitalization, while the pandemic has also accelerated the use of technology to facilitate the onboarding of employees hired during the pandemic. The results obtained align well with the previous findings stating that the automation of onboarding with AI reduces the burdens of various tasks on the HR department, making the entire process very efficient and convenient while enabling them to focus on more crucial tasks [63]. However, interestingly, it is found that Chinese firms have taken a step ahead and used artificial intelligence to even foster employee engagement by connecting new hires with other employees. Such use of artificial intelligence for onboarding has not yet been recognized by the Russian

and Indonesian firms. In both Russian and Indonesian firms, the use of these digital technologies and artificial intelligence has been limited to the paperwork processing with the provision of remote access to forms making onboarding time-efficient. On the other hand, in China's case, a more comprehensive use of digital technologies and artificial intelligence has been demonstrated by automating vital tasks such as account generation, orientation, training, and document generation, the use of AI-powered chatbots for guiding the employees and addressing their concerns, and for connecting the new hires with the existing employees for smooth onboarding using ONA. This shows that in Chinese firms, AI has been supporting socialization in the firms, fostering relationships among the employees, and enabling the newly hired to adjust well [64]. This also demonstrates that Chinese firms have realized the impact of demographic shifts on their onboarding process. As asserted by Troger [12], HRM must incorporate advanced digital technologies to adapt to the behavior of the digitally native younger population. Thus, technological tools have been found to be effective in motivating and retaining the new digitally forward workforce in organizations. Therefore, Chinese firms have effectively managed to transform the onboarding process for digitally native younger employees.

In the context of development, artificial intelligence is used in terms of performance management and appraisal, learning management of the employees, and employee benefits management. However, innovation is found to transform employee collaboration, engagement, and empowerment and contribute to employee retention. Thus, this fosters greater innovation owing to its ability to foster the increased exchange of ideas and information among the employees that enables them to contribute to greater creativity and innovativeness [66,67,69]. The results reveal that with the use of artificial intelligence, the firms can make more data-driven decisions for the performance management and appraisal of the employees, demonstrating the role of AI in the accurate assessment of employee performance [54]. Howbeit, the use of artificial intelligence in performance management is found to be more comprehensive in the case of Chinese and Russian firms as they have been using AI for forecasting potential performance and using adaptive learning models, while the performance management in Indonesian firms has been limited to performance evaluation only. Similarly, the use of AI has facilitated a more transparent process whereby HR managers and employees can track real-time performance and identify the issues in their performance. In the case of Chinese firms, the collection of the performance data from multiple sources such as supervisors, subordinates, and overseas subsidiaries with which employees have worked shows that performance management has been transformed to address the impacts of globalization as the performance data can be collected from other geographic subsidiaries as well. As a whole, the use of artificial intelligence makes the process free from being subjected to bias in the case of all 25 firms.

In the case of retention, the role of digital technologies, innovation, and artificial intelligence comes into play. Chinese firms again have taken a step ahead and have incorporated artificial intelligence for managing employee benefits. With these AI-powered digital technology and chatbots/programs, the employees are ensured that they are offered the best package that truly matches their needs, which contributes to the retention of the employees. Such application of AI at the retention stage was absent in the case of Russian and Indonesian firms, where the use of AI was limited to performance management and appraisal. Furthermore, all 55 firms have fostered innovation for the retention and development of employees by using platforms and co-working spaces. The use of these digital platforms has a direct influence on the retention and development of employees by fostering greater collaboration and engagement and improving employee mobility and knowledge exchange in the firms. The use of co-working spaces demonstrates an appropriate response to globalization and digitalization as employees can collaborate with one another regardless of their location, demonstrating greater employee mobility. Furthermore, the Chinese firms were also found to be actively using digital platforms for remote working to foster flexible working, which contributes to employee retention [8]. This demonstrates not only the impact of the COVID-19 pandemic on retention, whereby remote



working emerged as a norm, but also the demographic shifts, as the younger workforce appears to be motivated for greater flexibility and the incorporation of technology.

Finally, in the last stage of employee lifecycle management, exit/separation, the firms have incorporated digital technologies and artificial intelligence to transform the off-boarding process, which made it more automated and smoother. In all regions, the firms have used artificial intelligence for managing the paperwork, making it time-efficient and speedy. However, it is found that Chinese firms have taken a more comprehensive approach to the use of artificial intelligence in managing employees exists. Here, artificial intelligence is used for conducting an exit interview and for processing vital paperwork.

All in all, Chinese firms appear to have incorporated the most digital artificial intelligence innovation solutions to address the four megatrends: globalization, digitalization, COVID-19, and demographic shifts. The influence of the megatrend of demographic shifts was found to be absent in the case of both Russia and Indonesia, which have focused more on globalization, digitalization, and COVID-19. The following figure summarizes the impact of the megatrends on the employee lifecycle management of the three countries (Figure 6).

| | Artificial intelligence, digital technologies, and innovation | Globalization | COVID-19 | Demographic Shifts | Environmental and climate change |
|---|---|---|---|---|---|
| **China** | Strong, positive and negative | Strong and negative | Strong and negative | Strong and positive | Weak and positive |
| **Russia** | Strong and positive | Strong and negative | Strong and negative | Strong and negative | Weak and positive |
| **Indonesia** | Strong, positive and negative | Strong and negative | Strong and negative | Strong and negative | Weak and positive |

**Figure 6.** Impact of the megatrends on the employee lifecycle management of Chinese, Russian, and Indonesian companies.

Developing and using artificial intelligence and digital technologies have been proceeding at different paces in China, Russia, and Indonesia, particularly in their industrial sectors. Correspondingly, the impact of artificial intelligence and digital technologies was investigated with regard to China, the Russian Federation, and Indonesia. Increasing digitalization is expected to affect employment and HRM in all listed countries, particularly considering the consequences of the COVID-19-induced crisis, although it will show distinct features and different intensities between various industrial sectors and professional groups.

In the case of the Russian Federation and China, the state is central to administrating technological innovations, especially in the areas critical for national competitiveness. This affects business companies, including the resource market and labor market. However, technological innovations in Indonesia are mostly adopted by leading private companies with high competitiveness worldwide. Education and skills development at the macro level also determine the range of skills available in the labor market, leading to consequences for both companies and the workforce.

The wide spreading of digital transactions in China (such as the promotion of a cashless economy and transactions in WeChat) caused serious restructuring and cutting down of workplaces in the banking industry. The COVID-19 pandemic has accelerated robots' introduction in communal services and restaurants. Broader employment of machine learning technologies jeopardizes human employment itself and employment security in

particular, which clearly contradicts the principles of human-oriented HRM and sustainable development. Since the introduction of artificial intelligence, digital technologies are accelerating in the Russian Federation and Indonesia, and these countries risk facing a situation of labor market crisis. Globalization aggravates the labor division between countries, increasing overall productivity. Moreover, it increases interdependence among countries and vulnerability to partners' will. If relationships are damaged, and every party intends to harm the other, security matters are prioritized. Countries isolate themselves from each other, trading decreases, and each of them must rely only on itself, even regarding industries without a competitive advantage.

The COVID-19 pandemic caused massive staff dismissals (in every analyzed country) in multiple small companies engaged in particular sectors due to the absence of business activities. Furthermore, it has caused a swift demand for rising services in other sectors. This type of intersectoral temporal recruitment or worker assigning is a promising new model of workforce deployment strategy. Such crises as the COVID-19 pandemic inevitably enforces companies to reconsider their strategies of personnel management, in particular, regarding the search for ways to increase the flexibility of work offers, such as the cooperative use of personnel and alleviation of remote work.

However, despite all the advantages of the cooperative use of personnel model, including all benevolence that it caused during the pandemic, several issues of labor relationships have emerged that need close attention from regulatory authorities.

The influence of environmental matters (climate change, etc.) on HRM remains rather weak so far. This fact explained that the analyzed countries have only just started to participate in the enterprises of climate change and energy transition that are part of the agenda for sustainable development up to 2030. Generally, more and more countries aim to achieve zero-level emissions of greenhouse gases and develop appropriate national strategies. For example, China, at the end of 2020, announced the goal of achieving its peak of carbon emissions in 2030 and becoming carbon neutral in 2060. Such an ambitious plan requires the transformation of the existing industrial structure and energy consumption structure, as well as significant investments in the sector of renewable energy sources. The analyzed countries will need to introduce green HRM methods as a part of sustainable HRM systems for solving ecological problems. This includes HR strategies and practices that support the organization's goals for sustainable development [70–72]. At the moment, green HRM is not widely spread mainly because of the low priority given by companies and insufficient pressure on those companies themselves to introduce the practice of ecological management.

*5.2. Theoretical and Practical Contributions*

This research provides valuable insights into the ways in which Chinese, Russian, and Indonesian firms have incorporated digital technologies, artificial intelligence, and innovation into their employee lifecycle management. The study found that Chinese firms have adopted these tools in a more comprehensive manner compared to their Russian and Indonesian counterparts. Chinese firms are leveraging artificial intelligence, digital technologies, and innovation from each stage of the employee lifecycle, including attracting and recruiting employees, onboarding, performance management and appraisal, learning management, employee benefits management, and resigning employees.

One of the key findings of this research is that Chinese firms have fully automated their recruitment process with the use of artificial intelligence. This automation has resulted in increased effectiveness and accuracy of the recruitment process and has made it less worker-intensive and more cost- and time-efficient for the employers. The adoption of digital technologies and artificial intelligence has also enabled 25 companies to simplify their onboarding process with the elimination of various tasks. However, Chinese companies have taken this step further by using artificial intelligence to foster employee engagement by connecting new hires with other employees.

The use of artificial intelligence has also facilitated more data-driven decisions for performance management and appraisal of employees. Chinese and Russian firms have been found to be more comprehensive in their use of artificial intelligence for forecasting potential performance and using adaptive learning models, while the performance management of Indonesian firms has been limited to performance evaluation only.

Innovations have also transformed employee collaboration, engagement, and empowerment and contribute to employee retention. The use of digital tools and artificial intelligence fosters greater innovation by enabling the increased exchange of ideas and information among employees, contributing to greater creativity and innovation.

The practical contribution of this research lies in its ability to provide insights into the ways in which firms can incorporate digital technologies, artificial intelligence, and innovation into their employee lifecycle management. This is particularly relevant in the context of the megatrends of globalization, digitalization, demographic shifts, and the COVID-19 pandemic. The adoption of digital tools and artificial intelligence has become increasingly important as firms seek to navigate these trends and remain competitive.

One of the key practical implications of this research is that firms should consider fully automating their recruitment process with the use of artificial intelligence. This can increase the effectiveness and accuracy of the recruitment process and make it more cost- and time-efficient for the employers. The adoption of digital technologies and artificial intelligence can also simplify the onboarding process, enabling firms to focus on more crucial tasks. Furthermore, firms should consider using artificial intelligence for performance management and appraisal, as this can facilitate more data-driven decisions and improve the accuracy of performance assessments. This can also contribute to employee retention by fostering greater employee engagement and empowerment.

Another practical implication of this research is that firms should consider adopting innovative approaches to employee collaboration, engagement, and empowerment. The use of digital tools and artificial intelligence can facilitateagreater exchange of ideas and information among employees, contributing to greater creativity and innovativeness. This can also contribute to employee retention by fostering a positive organizational culture and a sense of belonging among employees.

Succinctly, this research provides valuable insights into the ways in which firms can incorporate digital technologies, artificial intelligence, and innovation into their employee lifecycle management. The adoption of these tools has become increasingly important in the context of the megatrends of globalization, digitalization, demographic shifts, and the COVID-19 pandemic. Firms that adopt these tools in a more comprehensive manner, such as Chinese firms, are likely to remain competitive and retain employees eventually. Therefore, firms should consider fully automating their recruitment process, simplifying the onboarding process, using artificial intelligence for performance management and appraisal, and adopting innovative approaches to employee collaboration, engagement, and empowerment.

## 6. Conclusions

### 6.1. Key Findings

To restate, the current study delineates the impact of artificial intelligence and innovations on employee lifecycle (ELC) management in the context of Russia, Indonesia, and China. The findings of the survey confirmed the propositions of the study and concluded that the adoption of innovation and artificial intelligence facilitates better management of each stage of the employee lifecycle at the companies. Thus, a significant relationship exists between the incorporation of artificial intelligence and innovation in organizations and the improved management of the ELC in organizations. This paper highlights that the incorporation of artificial intelligence in organizations transforms all the stages of the employee lifecycle, including recruitment, onboarding, retention, development, and exit. In particular, the adoption of artificial intelligence can lead to an automated, accurate recruitment and hiring process, eliminating the need for human intervention; a smoother

and easier onboarding process with smoother paperwork and documentation, continuous guidance, orientation, and training of the new employee, and better management of employee relationships; automated performance management; customized learning and training of the employees; automated enrollment and management of the employees' benefits; and automated paperwork and exit interviews.

Such benefits justify and rationalize the adoption of artificial intelligence for managing employee lifecycles in organizations. This supports the findings of a previous study that artificial intelligence facilitates the recruitment process, speeds up operations, automates various administrative tasks such as the management of employee performance, and provisions feedback on employee performance. Similarly, the incorporation of innovation in the organizations improves employee collaboration and management, develops a conducive environment that encourages employees to share ideas, develops a creative workforce, increases competitiveness, and cultivates the best experience of the employees. Thus, this study deduced that artificial intelligence and innovation pave the way for improved and enhanced employee lifecycle management in organizations.

### 6.2. Study Limitations

This study has some limitations. First, due to external factors, the scope of this study is limited to Russia, China, and Indonesia. The targeted audience of the study includes the organizations in which artificial intelligence and innovation are used for developing an ELM model. Although companies use numerous technologies and tools during ELM, this research has focused on three main factors, which are digital technology, innovation, and artificial intelligence. The reason behind the selection of these companies is determined by the increased use of digital technology, particularly artificial intelligence, in organizations in Russia, China, and Indonesia. Second, more extensive work must determine the typology of the level of digitalization of HR processes for each country. Third, this study did not focus on differences between gender, age, and positions in companies; we focused on respondents with the same level of education. Since no other characteristics were studied, participants in this study may not be representative of all companies as a whole. The methodology of this study needs to be replicated in larger groups of participants. In addition, based on such restrictive selection criteria for participants, the sample size was relatively small, which also led to poor representativeness of the results. Thus, this research serves only as a methodological study in this area.

This study's strengths are embedded in their theoretical and practical contributions. The findings of this study extended theoretical contributions to the extant literature by highlighting the link between technological innovations and advancements such as artificial intelligence, innovation, and better employee lifecycle management in organizations. Along with the theoretical contributions, the findings of this study also put forward practical implications for practitioners such as people managers, human resources managers, etc. The findings of this study can be used by practitioners to ensure efficient management of their employees' lifecycle, thereby being able to keep the employees satisfied and retained. This study has also encountered certain limitations that have affected the research findings. In this context, due to limited time and resources, only the data from limited participants were collected, which restricted the generalizability of the information to different industry and country settings.

### 6.3. Suggestions for Future Work

In recognition of the limitations identified in the previous section, it is recommended to future researchers that may intend to conduct an investigation in a similar area to conduct a study on a global scale by collecting data from various geographic regions. This would enable the researchers to produce results that can be generalized to all geographic contexts.

**Author Contributions:** Conceptualization, M.V.V. (Marina V. Vasiljeva) and H.X.; methodology, V.V.P. and J.L.; software, N.V.L.; validation, I.M.; formal analysis, A.O.Z.; investigation, I.D.E.; resources, H.X. and J.L. (regarding China), M.E.K. and D.Y.T. (regarding Russian Federation), I.M. (regarding Indonesia); data curation, H.X. and J.L. (regarding China), M.E.K. and D.Y.T. (regarding Russian Federation), I.M. (regarding Indonesia); writing—original draft preparation, H.X. and A.I.M.; writing—review and editing, S.Y.P.; visualization, M.V.V. (Maria V. Volkova); supervision, M.V.V. (Marina V. Vasiljeva); project administration, M.V.V. (Marina V. Vasiljeva). All authors have read and agreed to the published version of the manuscript.

**Funding:** This research received no external funding.

**Institutional Review Board Statement:** Ethical review and approval were waived for this study due to the reason of human involvement being limited to anonymous interviews.

**Informed Consent Statement:** Informed consent was obtained from all the subjects involved in the study.

**Data Availability Statement:** Not applicable.

**Conflicts of Interest:** The authors declare no conflict of interest.

## Appendix A

Interview Guide

This questionnaire has been designed to solicit information solely for research purpose. It is worth noting that this information will exclusively be used for academic purposes.

Please answer the following questions:

Name of Your Organization (it is not necessary to specify)_______________________

Your Position _______________________________

Your Gender Identity (it is not necessary to specify)_____________________

Your Age_____________________________________________________

Digital competency level

High_______________________________________________________

Medium _____________________________________________________

Low_______________________________________________________

Q1. Do you make use of artificial intelligence (AI) for recruitment and hiring in your firm?

Q2. How has automation in the hiring processes facilitated your organization?

Q3. Do you believe that the use of AI helps in the rightful selection of the target audience?

Q4. Apart from hiring and recruitment, how much does the integration of AI help in onboarding processes?

Q5. Is there any specific software that you use for collaborating within the organization?

Q6. Does AI help in the retention and development of employees?

Q7. How has AI shaped the performance management system within your organization?

Q8. What is the influence of AI integration on the retention of employees?

Q9. What is the role of AI in the exit process of employees?

Q10. Does AI, in general, help in the overall employee lifecycle?

Q11. How many people are employed in your company?

Q12. What is your company's scope of business?

Q13. Are you satisfied with the degree of automation of business processes in your company?

Q14. Are you satisfied with the degree of automation of human resource management in your company?

Q15. Do you know your company's digitalization index?

Q16. Does your company follow megatrends and technologies for the employees' development?

Q17. How hasCOVID-19 shaped the need for digital transformation?

Q18. Do you think that COVID-19 has affected the development of the employee life cycle? If yes, then how?

Q19. How can an innovation be implemented in the employee life cycle development?

Q20.  Please tell us about the type of megatrends and how they have affected your company's employees' life cycle development.

Thank you for your effort, time, and cooperation!

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
