# Peer review of "Sustainable Development of Employee Lifecycle Management in the Age of Global Challenges: Evidence from China, Russia, and Indonesia"

_sustainability, doi:10.3390/su15064987_

Round 1

Reviewer 1 Report

The paper I read was interesting and provided current evidence on long-discussed matters. However, I have some comments that need addressing towards improving it:

1.) The Introduction is way too long (7 pages). A lot of what exists there should be moved to the literature review section (section 2). In the introduction the authors should only provide a brief introduction to the theme of the study, its significance, the research questions, and what the reader is about to find by reading the rest of the paper (in a nutshell). All the rest of the detailed literature-led content should be moved in corresponding subsections of the literature review (chapter 2).

2.) The Hypotheses stated in lines 490 – 502 should be simplified. As its seems now, they look as if they have been “reverse engineered” – stated to fit the actual findings of the study, after the study was completed. The details within the hypotheses are not led by the literature (e.g. 100 people vs 10,000 people companies, etc.). You should check that the hypotheses be made simple and easy to understand, as well as easy to explain if they are supported or not by the findings.

3.) Section 3 should either be named methodology, or research methods. Not both (as it stands it is named “methodology and research methods).

4.) I don’t seem to understand Figure 2. The three hypotheses depicted are not identical to what the hypotheses were in lines 490-502 of the paper. You need to produce a figure that depicts a research model that is in concert with your hypotheses. Please revise as needed.

5.) In line 597 the authors mention semi-structured interviews. However, no interview guide is provided here. The authors should clearly state in text that their interview guide (e.g. list of questions employed during the semi-structured interviews) can be found in the Appendix. More importantly, they should provide theoretical evidence of how the interview guide was compiled. If the study guide was not based on evidence from the literature, the study loses much of its credibility.

6.) Section 3.1 seems to propose a model (figure 3) that has been composed based on the findings of the study (information collected from the cases enquired). However, it is placed within the methodology section. I believe it needs to be placed in the right spot within the next section (results) and the authors should explain better the process by which it was compiled through the information in their cases.

7.) Figure 6 is much larger (disproportionately) compared to Figures 4 & 5. Moreover, perhaps it would make more sense to have all three figures one under the other in one page, in order to compare the situation between countries.

8.) The onboarding process is depicted differently (in a table) than Figures 4,5,6 (Recruitment and hiring). There should be uniformity between the sections with regards to the reporting of results. I suggest adopting the format of Figures 4/5/6 for also the onboarding process and other processes that follow, as this format seems more easy to follow than the tabulated report of steps as a list (in Tables 1, 2, 3, 4).

9.) Section 4.6 is named “Evaluation of Structural model”. However, this research seems to have nothing to do with Structural Equation Modelling (SEM). It is qualitative work. Perhaps the authors here meant “Evaluation of Research Model”?? The section needs to be renamed.

10.) Table 5 needs to be significantly modified. The full hypotheses should be included here (all the text) so that the reader does not need to go back and forth to make sure what the hypothesis was. Also, it’s not enough to mention Yes/No in one column. You need to add two more columns to the table. One column should contain the supporting evidence from literature (copy some of the evidence from the background section in a bulleted format perhaps), The other column should contain the supporting evidence from the findings in this study (add some sample supporting evidence that you have mentioned in the previous subsections).

11.) The Discussion does not contain sections with regards to theoretical and practical contribution. They need to be added. How can your findings be utilized by practitioners in companies for instance (practical contribution), and how has existing theory been enriched by your findings (extra insight added to existing theory, proving/discarding past findings etc.). A couple of lines in the conclusion (section 6.2 “Strengths and Limitations”) speak of theoretical and practical contribution but they are firstly misplaced here and secondly, not enough. You need to dig deeper into your findings and combine them to build two sub-sections in the discussion that significantly explain the theoretical and practical contribution of your work.

12.) The study limitations (section 6.2) should not contain theoretical and practical contribution. They should instead provide a detailed account of the limitations of the study (study limited to three countries, limited according to sample size, non-experimental insight, no quantitative measurements combined with qualitative insight as of yet, and much more – you need to thoroughly re-visit this section and explain). The future research directions in section 6.3 should also be re-visited and re-edited based on the revised version of the study limitations section (once it has been completed). I suggest section 6.2 contains and is named “Study Limitations”, and section 6.3 accordingly contains and is named “Suggestions for Future Work”.

13.) A table with the demographic characteristics of the sample needs to be added in the results section. E.g., gender, age, position in company, etc. demographics collected from the participants of the study.

14.) English language is quite hard to follow in many places throughout the paper (especially expressions). The paper should definitely be proof-read by a native English speaker.

15.) The background section needs to be revisited to add more references (especially more recent ones). For example in Line 84: “A survey conducted by McKinsey Global Institute in November 2017”. McKinsey publisher reports yearly. Isn’t there a more recent study? 6 years have passed…

16.) In Line 149 the authors also note that: “evidence proposing that enhancing their environmental performance can positively benefit the businesses and organizations…These challenges are discussed in a number of studies, aside from [9].  à I suggest the authors add a more recent study that thoroughly discusses this matter and provides specific suggestions accordingly: (Kotsopoulos, D. Organizational Energy Conservation Matters in the Anthropocene. Energies 2022, 15, 8214. https://doi.org/10.3390/en15218214)

17.) The whole paragraph from line 84 to line 103 contains data from one single study (McKinsey 2017). More insight needs to be incorporated to justify the findings. Copying one study to that extent seems inappropriate (borderline plagiarism).

18.) In Line 121 the authors state: “A significant trend that has increased as a consequence of the recent COVID-19 pandemic is certainly in increased digitalization of the industrial chain globally”, and in Line 238-241: “The COVID-19 pandemic along with digital technologies and innovation have posed significant challenges to HRM, and this as a consequence has increased the need for a new set of skills for the workforce”. I suggest that the authors add recent studies explaining and supporting these facts thoroughly, e.g. by citing (among other studies): (Kotsopoulos, D., Karagianaki, A., & Baloutsos, S. (2022). The effect of human capital, innovation capacity, and Covid-19 crisis on Knowledge-Intensive Enterprises’ growth within a VC-driven innovation ecosystem. Journal of Business Research, 139(October 2021), 1177–1191. https://doi.org/10.1016/j.jbusres.2021.10.05)

19.) A lot of typing errors exist throughout the document. E.g. (but not limited to): a.) Line 933 & Line 1092: COVID-19pandemic à COVID-19 pandemic  (space missing between words), b.) Line 1094: COVID-19pandemichadcausedmassivestaffdismissals (all words stuck together – no spaces between them), c.) Line 1108: to2030 à to 2030, d.) Abstract  Lines 36-37:  the impact of the impact of megatrends (the impact is repeated twice), e.) Line 42: sustainabledevelopment ofEmployee (spaces missing between words). Please proof-read and correct similar issues as needed throughout the document.

Reviewer 2 Report

-Abstract is too long. Make it simple and firm. Add more significant findings.

- Introduction should be written in 3-4 paragraphs. The correlation between paragraphs is needed. 

- The discussion and analysis need to be discussed in more detail.  It lacks in correlation between sections. 

-  The conclusion should be summarized and tightened to reveal the overall finding. 

Reviewer 3 Report

The publication is interesting to read and covers important issues. The authors might consider to expand on the future research directions. This would guide other scholars on exploring based on the findings of this research and would better contribute to the future research in the field.

Author Response

Dear reviewer,

Thank you for the positive feedback. Indeed, we have in plans some broadening of our resarch scope in the future.

Sincerelu yours,

Xiang Hua et al.

Reviewer 4 Report

This paper decribes a comparison of China, Russia and Indonesia HR management policies for ensuring the sustainable development of employee lifecycle management in the age of global challenges. Various aspects of the automation using AI bots are included in the hiring loops in some countries and therefore, a comparison is necessary to comprehend the advantages and disadvantages of using AI in HR policies. The overall analysis concludes that digital transformation in HR policies can ensure sustainability in the long run as compared to manual operation. Also, the impact on climate change is yet to be correctly analyzed depending on the long term data trends in hiring policies.

In general, the paper seems good hwoever some improvements are necessary for the revised version.

1) Gaps are missing between the words in some phrases. This should be corrected for better understanding of the article.

2) English language improvements will even improve the quality of the current manuscript.

3) Comparison of HR policies of three countries Russia, China and Indonesia. What are the reasons to select these three countries? Is it possible to include one more country from west? maybe USA or at least a European country, maybe Germany or France for homogenous analysis of trends.

4) Digital transformation in HRM drived by the enhanced use of AI bots may pose challenges. For example:  

  • It may introduce machine-generated errors difficult to trace.
  • Perpetuating biases in hiring maybe observed.
  • It is possible to increased risks due to cybersecurity issues..... etc.

5) It would be good to include some reflections about these issues.

6) At least one figure with HRM data trends of the three countries can be included that can show the average trends in these countries based on data analytics.

Round 2

Reviewer 1 Report

I would like to congratulate the authors for the improvement of the paper. I have some further comments that I believe are necessary towards improvement for publication:

1.) The first part of chapter 2 (lines 130-188) would better be placed inside under a sub-chapter (i.e. add a subchapter title in the beginning of chapter 2 (e.g. entitled “Digital Technologies”). I also suggest that the authors add a few lines before that, explaining the sub-sections of the chapter (what the reader is about to read about). E.g. something in the lines of “In this section we analyze issues that affect HR, namely digital technologies, environmental changes,… (etc… based on the subchapters in this section)”.

2.) I believe that section 2.5. would better be placed right in the beginning of section 2 instead of the end.

3.) Since the authors have opted to remove the hypotheses in their study, they should also delete the two lines (486-487) that talk about hypotheses (“The following hypotheses were developed proceeding from the findings of the 486 extant literature and the objectives of this study:”).

4.) Line 154: stronglyinfluencing à strongly influencing

5.) Figure 1 looks odd as a number of steps are not methodological but simply descriptive. The authors should remove or revise it to better resemble a research model (a model that includes only the steps undertaken in the context of their research). Otherwise, they should produce a descriptive figure for their research that explains their approach (not named research model, but research approach). In the later case, they should carefully choose where arrows are needed, and where not.

6.) Line 970: Modelagainst à Model against

7.) The authors talk about “interviews” (e.g. in line 934), when in the methodology they outline questionnaires as the means for collecting evidence. I believe the best way is to describe what they have followed as “a structured interview employing a questionnaire that included free-form questions”. I suggest that the same description is referred to for the process they followed throughout the document (from the methodology to the discussion of results, etc.).

Reviewer 2 Report

Accept for publication 
